# Correlation of p53 oligomeric status and its subcellular localization in the presence of the AML-associated NPM mutant

Aleš Holoubek[1,☯], Dita Strachotová[2,☯], Kateřina Wolfová[1], Petra Otevřelova[1], Sára Belejová[2], Pavla Röselová[1], Aleš Benda[3], Barbora Brodská[1*], Petr Herman[2*]

**1** Department of Proteomics, Institute of Hematology and Blood Transfusion, Prague, Czech Republic,
**2** Faculty of Mathematics and Physics, Institute of Physics, Charles University, Prague, Czech Republic,
**3** Imaging Methods Core Facility at BIOCEV, Faculty of Science, Charles University, Vestec, Czech Republic

☯ These authors contributed equally to this work.
* Petr.Herman@matfyz.cuni.cz (PH); Barbora.Brodska@uhkt.cz (BB)

## Abstract

Tumor suppressor p53 is a key player in the cell response to DNA damage that suffers by frequent inactivating aberrations. Some of them disturb p53 oligomerization and influence cell decision between proliferation, growth arrest and apoptosis. Active p53 resides mostly in the nucleus, degradation occurs in the cytoplasm. Acute myeloid leukemia (AML)-related mutation of NPM (NPMmut) induces massive mislocalization of p53 to the cytoplasm, which might be related to leukemia initiation. Since both proteins interact and execute their function as oligomers, we investigated the role of perturbed p53 oligomerization in the p53 mislocalization process in live cells by FLIM (fluorescence lifetime imaging microscopy), fluorescence anisotropy imaging (FAIM), fluorescence cross-correlation spectroscopy (FCCS) and immunochemical methods. On a set of fluorescently labeled p53 variants, monomeric R337G and L344P, dimeric L344A, and multimeric D352G and A353S, we correlated their cellular localization, oligomerization and interaction with NPMmut. Interplay between nuclear export signal (NES) and nuclear localization signal (NLS) of p53 was investigated as well. While NLS was found critical for the nuclear p53 localization, NES plays less significant role. We observed cytoplasmic translocation only for multimeric A353S variant with sufficient stability and strong interaction with NPMmut. Less stable multimer D352G and L344A dimer were not translocated, monomeric p53 variants always resided in the nucleus independently of the presence of NPMmut and NES intactness. Oligomeric state of NPMmut is not required for p53 translocation, which happens also in the presence of the nonoligomerizing NPMmut variant. The prominent structural and functional role of the R337 residue is shown.

**Data availability statement:** All raw images from western blots are available from the Figshare data repository under a DOI: https://doi.org/10.6084/m9.figshare.28667498.

**Funding:** The work was supported by the Czech Science Foundation – grant No 22-03875S. BB was also supported by MH CZ – DRO (IHBT, 00023736), the project for conceptual development of the research organization. AB acknowledges Imaging Methods Core Facility at BIOCEV, an institution supported by the MEYS CR (LM2023050 Czech-BioImaging). The funders had no role in study design, data collection and analysis, decision to publish, or preparation of the manuscript. There was no additional external funding received for this study.

**Competing interests:** The authors have declared that no competing interests exist.

## Introduction

Tumor suppressor p53 is a nuclear phosphoprotein encoded by the *TP53* gene. It plays its primary role as a transcription factor in the cell response to DNA damage where it induces growth arrest or apoptosis [1,2], mostly via transcriptional activation of genes involved in the regulation of the cell cycle [3–5]. The level of p53 protein in intact cells is tightly regulated and kept low by a rapid turnover, with p53 having a half-life typically tens of minutes [6,7]. *TP53* is the most commonly mutated gene in human cancers and *TP53* germline mutations are responsible for the cancer-prone Li-Fraumeni syndrome [8,9]. The function of p53 protein can be inactivated through knock-out, mutation, posttranslational modifications, and interaction with cellular or viral proteins, which can lead to its mislocalization, e.g., to the cytoplasm. The p53 inactivation is a key step in over half of all human cancers (see reviews [10–12]), although the frequency and localization of *TP53* alterations vary among cancer types [9]. Next to its canonical function as the transcription factor, p53 was shown to play additional roles also in many other processes connected with tumorigenesis and tumor suppression (reviewed by Vousden and Prives [13]). Specifically, its direct interaction with both pro- and antiapoptotic proteins from the Bcl-2 family on the mitochondrial membrane contributes to the regulation of apoptosis [14,15].

The p53 is a modular protein (Fig 1), consisting of the N-terminal activation domain, the central DNA-binding domain (DBD), the tetramerization domain (Tet), and the C-terminal regulatory domain [11,12]. Tumor suppressor p53 missense mutations are the most frequent genetic alterations in human cancers [16], the majority of mutations have been found in the DBD and led to deficiency in the sequence-specific DNA-binding [17]. Until recently relatively few tumor-derived mutations have been reported for the Tet domain (326–356 AA) [18], where inhibition of the tetramerization results in the inhibition of the protein function. Nevertheless, Choe et al. [19] reported a similar mutation frequency for germline mutations in Tet and DBD. Point mutations in the oligomerization domain do not necessarily lead to a dysfunctional p53 protein [20,21]. According to Ishioka and co-workers [22], specific point mutations in the oligomerization domain of p53 rendered it ineffective for certain p53 functions. However, not all tested p53 functions were disabled for each particular mutant. The transcriptional activity of several p53 variants mutated in the Tet domain is summarized in Gencel-Augusto et al. [18]. Similarly, a set of recently published results on A347D mutant summarizes inhibiting effect of this mutation on the p53 transcription activity and simultaneous induction of the neomorphic function in apoptosis [19,23,24].

Mutations in the Tet domain can result in oligomerization defects with possible clinical consequences, as demonstrated by certain missense pathogenic germline *TP53* mutations identified in the Li-Fraumeni syndrome [8,20]. In case of L344P, the resulting monomeric mutant is completely inactive [25] and its DNA binding activity is markedly reduced [26]. On the other hand, R337C can form homo-tetramers, however, only about half of the protein molecules associate in oligomers at physiological conditions [8]. Another consequence of mutations in the Tet domain is an altered interaction of p53 with other proteins. For example, Lomax et al. [20] have shown that both L344P and R337C have reduced ability to bind to MDM2 which is the main

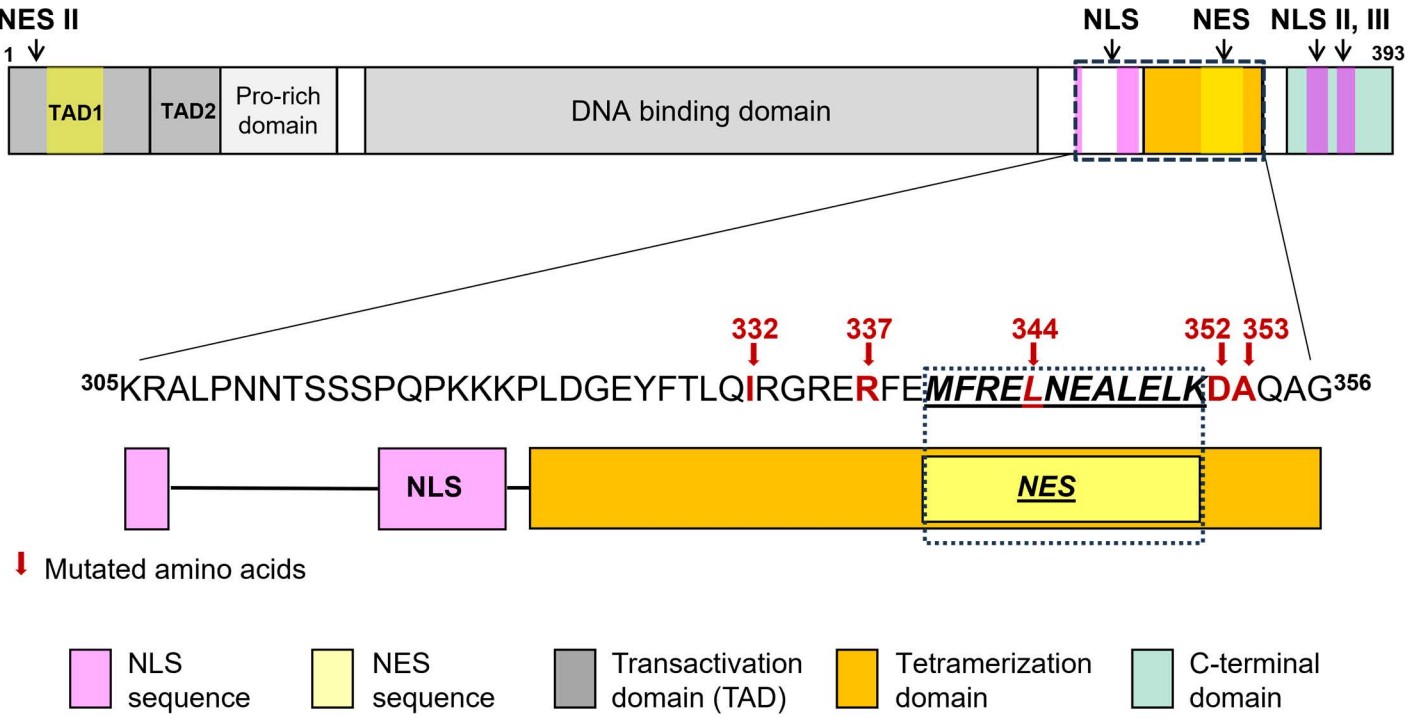

**Fig 1. Schematic representation of p53 protein domains with indicated positions of the NES and NLS sequences.** The amino acids mutated in this study are in red and marked with an arrow.

regulator of the p53 level [27]. Garcia-Cano et al. [28] showed that specific modifications enabling the p53 interaction with DNA require association of p53 in tetramers. However, correlation of the mutations with oligomerization and interaction ability remains to be elucidated. Frequent mutations in R337 residue have been reported to greatly impact p53 functionality. In particular, germline mutations R337C and R337H are widely detected in cancers [29,30]. The DNA binding activity assay of R337C showed high binding affinity to DNA but only a partial activation of the reporter gene. This suggests that mere binding to DNA does not necessarily affect the transcription of the downstream gene. Recently, Rigoli et al. [31] showed partial rescue of p53 transactivation potential, when R337D mutation was compensated by D352R leading to re-formation of salt bridge between these two residues.

## Oligomeric states of p53

The p53 tumor suppressor is critical for genesis of many cancers, including oncohematological diseases. In live cells, the protein appears in the form of monomers, dimers and tetramers [18]. Defective p53 oligomerization influences the cell decision between proliferation, growth arrest and apoptosis [25]. While tetramers trigger apoptosis and cell death, dimers retain only the growth arrest function. *In vitro* fluorescence correlation spectroscopy (FCS) suggests that the main oligomeric state of p53 in resting cells is a dimer [32]. This agrees with Gaglia et al. [33], who found that p53 is under basal conditions present mainly in dimers (~59%), followed by monomers (~28%), and tetramers (~13%). The *in vitro* association of dimers into tetramers has been reported as a slow process, which suggests that it requires additional players and/or modifications [32,34]. The main p53 function, the extensive transcription regulation, is attributed to p53 tetramers bound to the DNA [35]. Tetramerization is a process that should be facilitated by increased p53 concentration in the impaired cell [36–38], whereas in intact cells, low level of p53 is maintained by E3 ubiquitin ligase Mdm2. Nevertheless, functional tetramers were reported to be formed prior the increase of p53 levels upon DNA damage [39]. This suggests that other

factors, e.g., shift in the equilibrium between oligomeric variants by protein-protein interactions or p53 modifications, contribute to the p53 tetramerization. According to this concept, more than 90% of p53 was observed in tetramers after Neocarzinostatin-induced DNA damage without detection of substantially elevated protein levels [33]. There is a large body of evidence that mutations leading to a total loss of p53 oligomerization, i.e., coding for the monomeric mutants, possess high cancer-inducing potential and cause loss of p53 function [18,40,41]. However, different conclusions resulted from experiments with p53 mutants with partially impaired oligomerization. While higher stability of dimer-forming mutants was found by some researchers [25], others reported that the Mdm2-mediated p53 cytoplasmic export and subsequent degradation happens preferentially for the dimeric p53 mutants [42].

Tetramerization of p53 extremely increases its specific binding to appropriate DNA sequence *in vitro* – 1000-fold compared to the monomer and 6-fold compared to the dimer [35]. Nevertheless, p53 tetramerization can cause also lowered thermal stability of the complex at physiological temperatures compared to its dimeric (L344A) and monomeric (L344P) form [43], probably due to a crosstalk of DBDs in the tetramers.

## Subcellular localization of p53

The function of p53 in regulatory pathways is tightly connected to its intracellular localization. Generally, the active form is stabilized in the nucleus, the degradation is linked to translocation of ubiquitinylated protein to the cytoplasm. The protein possesses several well-defined localization sequences: three nuclear localization sequences (NLS), one highly conserved leucine-rich nuclear export signal (NES) in its C terminus [44], and the other NES II located in its N terminus [45], (Fig 1). NLS and NES mediate shuttling of p53 between the nucleus and the cytoplasm, which allows a dynamic regulation of its level [46].

The main p53 NLS is bipartite and includes basic amino acids at positions 305 and 306, and between positions 316–322 [47,48]. Two additional nuclear localization motives, NLS II and NLS III at positions 366–372 and 376–381, respectively, were evaluated as weaker [47]. Importin-α3 works as a p53-specific adaptor in the nuclear import machinery [49].

The main p53 NES comprises residues 340–351. It lies within the p53 tetramerization domain, which is composed of a β-strand at residues 326–333, and an α-helix at residues 335–356. The two β-strands of adjacent p53 molecules can align during formation of dimers, which allows combination of the two dimers in a tetramer [50–52]. Since three of the key hydrophobic residues of the NES (L344, L348 and L350) lie at the dimer interface, they mediate also the tetramer formation [53], linking thus mutations in the NES to aberrations of both the oligomerization status and the export rate. Nuclear export of p53 by exportin-1 (XPO1) is promoted, e.g., by Mdm2-mediated ubiquitination [54], which is believed to expose the C-terminal NES [55].

Gradient between mobile fractions of p53 in the cytoplasm and nucleus is given by the equilibrium between the nuclear export and import rates mediated by the NES and NLS signals. Turnover of p53 in the cell is regulated by interaction with factors such as Mdm2, p14Arf or NPM, which participate in a dynamic equilibrium loop regulating its degradation [56–58]. According to the structural analysis, NES is hidden when p53 persists in the tetrameric state [44]. Under the stress stimuli, the tetramer population becomes enriched and the NES masked. The activated transcription factor is therefore supposed to be kept inside the nucleus where it orchestrates cellular stress response. The generally accepted model states that if the NES is exposed on the surface of the p53 molecule (such as for monomeric p53, or upon binding of another protein, e.g., ARF [59]), cytoplasmic localization of p53 is promoted. Liang and Clarke [60] hypothesized that the p53 region between AA 326 and 355 could inhibit the nuclear import of p53 by masking the NLS and blocking the binding site for importin α.

Other factors affecting the localization of p53 are specific conformations and modifications of the protein. According to Nie et al. [55], both conformational change and ubiquitination induced by Mdm2 binding to p53 are necessary for its nuclear export. Presence of the two independent NES sequences could provide cells with greater flexibility to regulate p53 localization. Phosphorylation of serines in the N-terminal NES has a stabilization function and sequesters p53 in the

nucleus [45]. SUMOylation of p53 at Lys386 leads - on the other hand - to its increased cytoplasmic localization [61]. In addition, p53 localization is strongly affected also by binding to its interaction partners, e.g., to mostly immobile DNA or mobile nucleophosmin (NPM) [62].

## Nucleophosmin and its interaction with p53: an oligomer issue

NPM is an abundant, multifunctional, and mostly nucleolar protein occurring in cells prevalently in pentamers [63]. While the NPM oligomerization is mediated by its N-terminal domain, the nucleolar localization is ensured by the nucleolar localization sequence (NoLS) near the C-terminus (S1 Fig [64]). Characteristic C-terminal mutation of the *NPM1* gene resulting in the aberrant localization of mutated NPM protein (NPMmut) to the cytoplasm appears in about 50% of acute myeloid leukemia (AML) with normal karyotype [63,65]. As we have previously shown [66], NPMmut also forms oligomers in the cytoplasm. According to Colombo et al. [67], NPM interacts directly with p53. Recently we proved that AML-related NPM mutation does not impair this interaction and NPMmut causes p53 delocalization into the cytoplasm, which may possibly affect the p53-driven stress response [62]. Two regions in p53, AA 175–196 and 343–363, were identified as important for interaction with NPM [68]. Since then, no study has addressed this issue and no data exist on the oligomeric state of the individual proteins during this interaction. Since both NPM and p53 execute most of their cellular functions as oligomers, it is important to check the relation of the oligomeric state of p53 and its AML-associated phenotype. Although p53 mutations are relatively scarce in AML, non-mutational p53 abnormalities (e.g., altered post-translational modifications, p53 expression level or subcellular localization) appear to be rather frequent in AML [69]. The NPMmut-induced alteration in p53 localization thus might be one of the leukemia initiating factors. Since the role and consequences of the perturbed oligomerization associated with p53 abnormalities in the AML initiation is mostly unknown, we focused on the mechanism of p53 being pulled-out by NPMmut.

## Materials and methods

### Cell culture and chemicals

Human embryonic kidney cell line 293T (HEK-293T) was kindly provided by dr. Němečková (IHBT, Prague). Cells were maintained at DMEM-F12 (LM-D1223, Biosera) supplemented with 10% FBS (FB 1090/500, Biosera) and antibiotics (100 U/ml penicillin, and 100 µg/ml streptomycin, P4333, Merck) in a humidified atmosphere at 37°C/5% $CO_2$.

### Plasmid construction and cloning

Preparation of plasmids for production of NPMmut labeled with Cerulean or mVenus is described in Strachotova et al. [70], plasmid ensuring production of C_Δ117NPMmut is constructed for this study similarly using appropriate Fw primer (S1 Table).

DNA fragment with NowGFP sequence was subcloned to Cerulean-C1 plasmid, a gift from M. Davidson & D. Piston (Addgene plasmid # 54604 [71]). Sequence for Cerulean was replaced with NowGFP sequence that was PCR-amplified from NowGFP/pQE-30 using extended primers containing XbaI and XhoI (Thermo Scientific) restriction sites. The NowGFP/pQE-30 plasmid was a gift from K. Lukyanov (Addgene plasmid; #74749 [72]). The fragments containing the NowGFP sequence were then ligated to vector prepared from the Cerulean-C1 plasmid by excision with restriction enzymes (Thermo Scientific) between NheI and XhoI sites (S1 Table). The resulting pNowGFP-C1 plasmid was used for further molecular cloning.

Plasmids for producing fluorescently labeled p53 variants were constructed on the original pmRFP1-C2:p53wt [62] and pmVenus-C1:p53wt plasmids [70]. DNA fragment corresponding to specific *TP53* transcript, isoform a (RefSeq. NM 000546.6, from NCBI database), was PCR-amplified from the pmRFP1-C2:p53wt plasmid using appropriate extended primers (S1 Table). Subsequently, it was sub-cloned to the vector pNowGFP-C1 using XhoI and PstI unique restriction sites (Thermo Scientific) and T4 DNA ligase (M0202S, NEB). *TP53* sequence alterations leading to p53 mutations were

introduced to the pNowGFP-C1:p53wt construct by Q5 Site-Directed mutagenesis kit (E0554S, NEB) using appropriately designed plasmids (S1 Table). Afterwards, DNA fragments with the altered sequences were again PCR-amplified using extended primers (S1 Table) and reversely subcloned back to the original pmRFP1-C2 and pmVenus-C1 vectors [70]. The classic bipartite main NLS "KR-X$_{10}$-KKK" of p53 was inhibited by changing KKK for AAA in the plasmids constructed for production of mRFP1-labeled p53 variants using the Q5 Site-Directed mutagenesis kit (NEB) and appropriately designed plasmids (S1 Table). All constructed plasmids were amplified in TOP10 *E. coli* competent cells (ThermoFisher Scientific), then purified with the PureYield Plasmid Miniprep System (A1223, Promega). All introduced and mutated constructs were verified by sequencing. List of constructs can be found in S1 Table.

### Cell transfection

HEK-293T cells were seeded to the cell density of $1 \times 10^5$/ml 24h prior to transfection. Transfection was achieved with jet-Prime transfection reagent (#101000046, Polyplus transfection) following the manufacturer's protocol. The growth medium was replaced 4h after the transfection, and cells were further grown for 20–40 hours before analysis.

### Glutaraldehyde crosslinking

Transfected cells were grown to 80% confluence and harvested in Hepes lysis buffer (50mM Hepes, 150mM NaCl, 1% NP-40, 5mM EDTA, pH 7.4) with freshly added protease inhibitor cocktail (P8340, Merck). Glutaraldehyde (GA) was added to 0.025% final concentration and lysates were rotated for 30min at RT. Finally the samples were mixed 1:1 with 2x concentrated sample buffer (SB, 100mM Tris.HCl pH 6.8, 4% SDS, 200mM DTT, 20% glycerol), incubated at 95°C for 5min, and stored at -20°C until used for SDS-PAGE. Control samples were prepared by the same procedure without GA addition.

### Immunoprecipitation

Co-immunoprecipitation was performed as described earlier [73]. Briefly, transfected cells expressing fluorescent proteins were processed after 40h-incubation. GFP-, RFP- and DYKDDDDK-Trap_A systems (gta-20, rta-20, ffa-20, Proteintech) were used following the manufacturer's instructions. Thanks to the structural similarity of Cerulean, mVenus and NowGFP, the GFP-Trap system was used for precipitation of all these tags. FP-expressing adherent cells were washed with ice-cold PBS and scraped from the dish. The cell pellet was lysed in the lysis buffer (LB, 10mM Tris.HCl pH7.5, 150mM NaCl, 0.5mM EDTA, 0.5% NP-40, protease and phosphatase inhibitors) on ice for 30min and centrifuged at 20000g/10min/4°C. The lysate was applied on Trap_A beads pre-incubated in washing buffer (WB, 10mM Tris.HCl pH7.5, 150mM NaCl, 0.5mM EDTA) and rotated for 1h at 4°C. Then the beads were pelleted and extensively washed in the WB, resuspended in 2xSB, heated for 10min, and centrifuged at 2500g/2min/4°C. Supernatant was stored at -20°C until used for SDS-PAGE. At least 3 replicates per experiment were performed.

### Seminative SDS-PAGE

Transfected cells were grown to 80% confluence, harvested in LB on ice for 30min and centrifuged at 20000g/10min/4°C. The lysates were mixed 1:1 with 2x concentrated native sample buffer (100mM Tris.HCl pH 6.8, 20mM DTT, 20% glycerol) and stored at -20°C until used. SDS-PAGE was then performed with 0%-0.1% SDS concentration in both the 8% acryl-amide gel and electrophoresis buffer.

### Immunoblotting

Five to ten microliters of each sample were subjected to SDS-PAGE and transferred into PVDF membrane (#1704275, BioRad). Mouse monoclonal antibodies against β-Actin (sc-47778), p53 (sc-126), GFP (sc-9996), dsRed (sc-390909), and

NPM (sc-70392) were from Santa Cruz Biotechnology. All mouse primary antibodies were used at a dilution 1:100–1:500. Rabbit monoclonal antibody against NPMmut from Covalab (pab50321), mouse monoclonal antibody against NPMwt C-terminus from Abcam (ab10530) and mouse monoclonal anti-DDDDK from Exbio (#11–425-C100) were used at 1:1000 dilution. Anti-mouse and anti-rabbit HRP-conjugated secondary antibodies were purchased from Thermo Scientific (#31430 and #31460) and used at concentrations 1:10000–1:50000. ECL Plus Western Blotting Detection System (#28980926, GE Healthcare) or SuperSignal West Atto Ultimate Sensitivity Substrate (#A38556, ThermoFisher Sci) were used for chemiluminescence detection and signal was evaluated by G-box Chemi XX6 digital imaging device (Syngene Europe). Images included in Figures are always representative of at least three independent experiments.

## Live-cell imaging

Cells were grown on a glass bottom Petri dish (D29-14–1,5-N, Cellvis). The subcellular distribution and co-localization of FP-fused protein variants were observed under the confocal laser scanning microscope FV1000 (Olympus Corporation) with UPLSAPO 60x NA 1.35 oil immersion objective (Olympus). Data were processed with the FluoView FV10-ASW v3.1 software. Cerulean, mVenus and mRFP1 emission was sequentially excited at 405, 488 and 543 nm, respectively, using the DM405/488/543/647 dichroic mirror at the excitation path. Cerulean emission was collected with the BA430–490 bandpass filter, mVenus and mRFP1 emission was isolated by the FV12-MHBY filter cube in the emission path. Live-cell imaging was done at least in triplicates.

## FLIM and anisotropy imaging

Fluorescence lifetime imaging and data analysis was performed as described previously [74]. Briefly, experiments were performed using inverted IX83 confocal microscope with FV1200 scanner (Olympus, Tokyo, Japan), cell-cultivation chamber (Okolab, Pozzuoli, NA, Italy), and FLIM add-ons (Picoquant, Germany) with picosecond semiconductor lasers, cooled GaAsP detectors and TimeHarp 260PICO TCSPC detection (all PicoQuant, Berlin, Germany). Pulsed 488 nm excitation (LDH-DC-458) was directed to the sample via a polarization-maintaining optical fiber, DM405/488 dichroic mirror (Olympus) and the UPLSAPO 60XW NA = 1.2 water-immersion objective (Olympus). To avoid pile-up artifacts, the data collection rate at brightest pixels was kept below 5% of the laser repetition frequency [75].

For the anisotropy imaging, the emission from the sample was split to the parallel and perpendicular polarization components (relative to the excitation polarization) by a polarization beam splitter placed in the emission path. The components were spectrally filtered by a pair of the 534/30 bandpass filters (Semrock) before detection. To minimize depolarization artifacts, UPLSAPO30XS oil immersion objective (Olympus) with a reduced numerical aperture (NA = 1.05) was used [76]. G-factor normalizing transmission efficiency of the two polarized detection channels [77] was measured using fast-rotating molecules of rhodamine G in ethanol, which should exhibit zero fluorescence anisotropy after very fast initial depolarization. Then steady-state emission anisotropy was calculated for each pixel of the image [78].

All experiments were performed at 37°C. Both FLIM and anisotropy data were processed using the SYMPHOTIME software (Picoquant).

## Fluorescence bleaching

In the control FRET experiments the acceptor was fully photobleached in the selected ROI by the 562 nm laser. Presence of FRET was evaluated from the increase in the donor emission lifetime after the acceptor photodestruction caused by the disruption of the FRET channel.

In analogy, in the control homoFRET experiments the intensity of the fluorescent tags was photobleached to 30% of its initial value using intense 488 nm laser in order to reduce the effective fraction of labeled p53 in the oligomers. Since homoFRET in the p53 complexes manifests itself by fluorescence depolarization, bleaching results in the anisotropy increase, when oligomers are present in the sample.

**PIE-FCCS on live cells**

Fluorescence Cross-Correlation Spectroscopy with pulse interleaved excitation (PIE-FCCS) was performed on the Abberior Instruments Expert Line STED system built around a Nikon Eclipse Ti-E microscope body, extended with the Time Harp 260P module (PicoQuant) and a top-stage incubator (Okolab), located at Imaging Methods Core Facility, BIOCEV, Vestec. For measurements a Nikon Plan Apo IR 60x NA = 1.27 water immersion objective was used. The sample was illuminated with pulsed 485 nm and 561 nm lasers, each pulsing at 20 MHz, and the pulse trains shifted by 25 ns. Laser powers at the sample plane were 1.5 µW and 3.9 µW, respectively. Perfect spatial overlay of the laser beams was checked before every measurement using TetraSpeck beads and the size of the confocal volume was calibrated by measuring FCS for Atto488 dye in water with a known diffusion coefficient 400 µm²/s at 25°C. The fluorescence signal passing the pinhole (set to 0.91 AU for 640 nm emission) was filtered for channels A and B by 500–550 nm and 580–630 nm emission filters, respectively, detected by single-photon counting detectors (Excelitas Technologies, Waltham, MA, USA) and stored using TTTR data format.

Measurements were done at 37°C. Cells with a typical signal distribution showing low expression of fluorescent proteins were selected by a confocal scanning. Then the laser beam was parked into the nucleus and 60s long data acquisition was taken. The TTTR data were processed using a custom-written software "TTTR Data Analysis" [79]. Auto- and cross- correlation functions were calculated from the spectral cross-talk free PIE data and fitted using standard one component 3D-diffusion model assuming Gaussian shape of the detection volume described by:

$$G(t) = 1 + \frac{1}{N_p} \left( \frac{1}{1 + t/\tau_D} \right) \left( \frac{1}{(1 + t/\tau_D)^{1/2}} \right),$$

(1)

where $N_p$ is a number of emitting particles in the measured volume and $\tau_D$ is related to the translational diffusion coefficient D and radius of the confocal detection volume ω by:

$$\tau_D = \frac{\omega^2}{4D}.$$

(2)

Due to the low excitation powers applied, no components accounting for photophysical processes were needed and used. The cross-correlation between the two fluorescence signals was evaluated by the cross-correlation amount Q defined as [80]:

$$Q = \max \left[ \frac{G_{0_x}}{G_{0_B}} \; ; \; \frac{G_{0_x}}{G_{0_A}} \right],$$

(3)

where $G_{0_x}$, $G_{0_A}$, and $G_{0_B}$ are initial values of the fitted cross correlation function, autocorrelation function of mVenus in the channel A and mRFP1 in the channel B, respectively. This quantity increases with the increasing fraction of the two-color complexes containing red and green monomers diffusing together across the measurement volume.

**Statistical evaluation**

Statistical evaluations were done using GraphPad Prism software version 10.2.3 and the MATLAB software package. Non-parametric ANOVA test with the Tukey-Kramer post analysis was used to compare individual sample pairs. The p-value <0.05 was considered as a significant difference. 95% confidence graphs of differences between individual pair of samples are shown together with the statistical evaluation graphs. Appropriate p-values for all statistics are specified in S2 Table.

## Results

### Co-transport of p53 oligomerization variants with NPMmut to the cytoplasm

As a shuttling protein, p53 is constantly transported from the nucleus to the cytoplasm and *vice versa.* In the presence of an AML-related NPM mutant (NPMmut), which is aberrantly localized in the cytoplasm, the cytoplasmic fraction of p53wt significantly increases due to the NPMmut-p53wt interaction [62]. The resulting aberrant localization of p53 may result in an inappropriate stress response of the cell.

In order to investigate relation between p53 delocalization and its oligomerization, we prepared a set of p53 oligomerization variants for exogenous expression. First, p53 was mutated in the Tet domain within the main NES to obtain monomeric (L344P) and dimeric (L344A) variants [8,35]. Since the compromised NES could affect co-transport of the complex, we introduced also alternative oligomerization-impairing mutations [81] in sites adjacent to the main NES, specifically one monomeric (R337G) and two dimeric (D352G, A353S) ones. Then, HEK-293T cells were co-transfected always with one of the p53 variants and NPMwt or NPMmut, and the subcellular localization was analyzed (Figs 2 and S2). As shown in S2 Fig (D), the expression levels of exogenous NPMwt and NPMmut are comparable. All the p53 variants are localized in the nucleus when co-transfected with NPMwt (S2 Fig). While p53wt is pulled-out by NPMmut (Fig 2A), the monomeric L344P and R337G (Fig 2B) remain localized in the nucleus in the presence of NPMmut. Absence of the cytoplasmic signal of the monomeric variants indicates a lack of interaction with NPMmut. From the putative dimeric mutants (L344A, D352G, and A353S), only A353S was translocated to the cytoplasm similarly to p53wt, when co-expressed with NPMmut. L344A and D352G constructs remained in the nucleus (Fig 2C). Inconsistent results in case of putative dimeric mutants suggest that NPMmut-induced p53 delocalization might not be a simple function of the p53 oligomerization state.

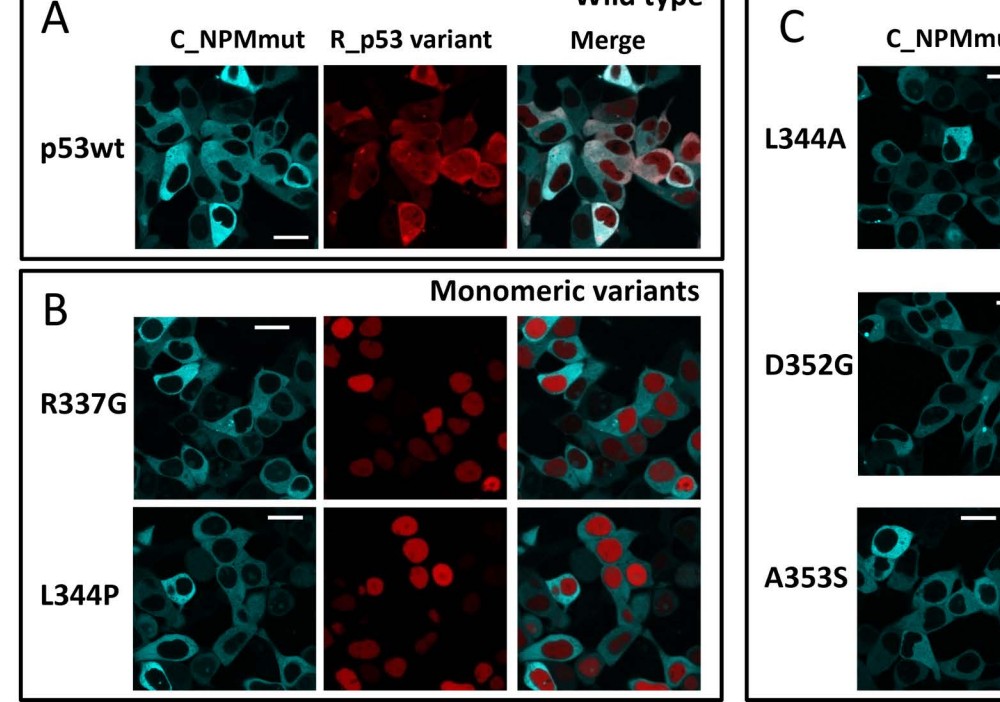
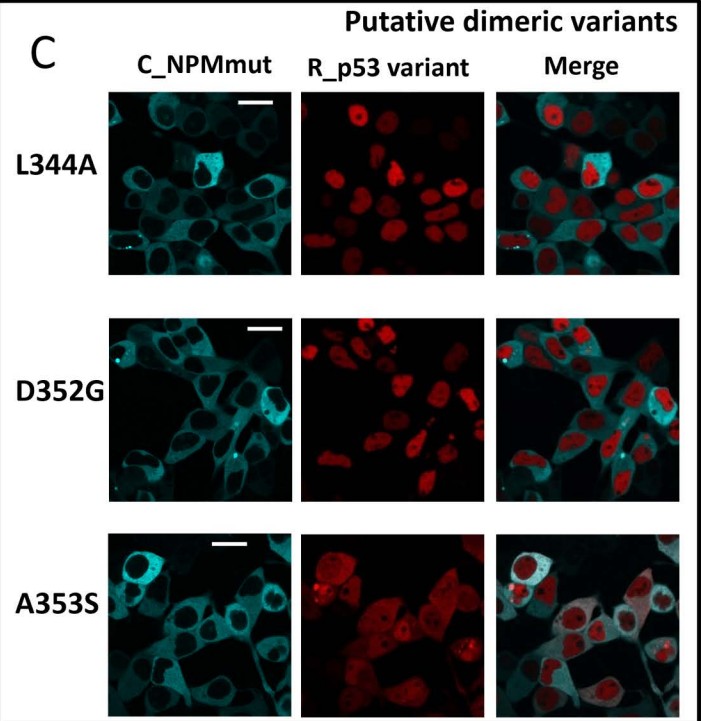

**Fig 2. Cellular localization of mRFP1-labeled variants of p53 (R_p53) co-expressed with Cerulean labeled NPMmut (C_NPMmut).** A) p53wt, B) monomeric variants, and C) putative dimeric variants. Bar represents 20 µm.

## Verification of the oligomerization state of p53 mutants

To verify the predicted oligomerization state of the used p53 variants, we examined them by Western blotting of cell lysates with native interaction complexes being conserved by glutaraldehyde crosslinking. This experiment should distinguish dimeric and tetrameric variants of fluorescently labeled p53, Fig 3. We can see that while p53wt is present mainly in tetramers, a prevailing monomeric form is clearly seen for the L344P and R337G constructs. Among the non-monomeric mutants, only L344A predominantly forms dimers, while D352G and A353S gel patterns are not dimeric and resemble the oligomeric one of p53wt. The experiment was repeated with various tags with the same results, S3 Fig. This apparent disagreement between the proposed oligomerization state and the experimental *in vitro* results led us to investigate more deeply the oligomerization of the p53 constructs in live cells.

## Oligomerization status of p53 mutants in live cells

Several independent approaches were chosen to characterize the oligomeric state of the p53 mutants in live cells. Specifically, homo and heteroFRET imaging in the cells expressing fluorescently tagged p53 was complemented by fluorescence correlation spectroscopy for characterization of the oligomerization at low p53 concentrations.

Results of the live-cell FLIM imaging are shown in Fig 4. For the FLIM-FRET measurements, all protomers of p53 mutants were labeled either with mVenus (donor) or mRFP1 (acceptor). Both color variants were co-expressed in cells and a formation of mixed complexes was examined. The presence of FRET was tested by the acceptor bleaching, when increase in the donor lifetime upon the acceptor destruction indicates the presence of FRET in p53 complexes. In Fig 4A we can clearly see that FRET was detected for p53wt and L344A, D352G, and A353S mutants. Conversely, oligomer formation is unlikely for R337G and L344P variants, since FRET-induced lifetime changes were much smaller and contrasted with the values obtained for the oligomeric variants. The statistical analysis is presented in Fig 4B. Data suggest that in live cells R337G and L344P are mainly in the monomeric state, while L344A, D352G, A353S, and p53wt form higher oligomers. This agrees with the results from immunoblots (Fig 3).

Steady-state fluorescence anisotropy imaging [78] served as an additional independent verification of the heteroFRET data. Photobleaching of the fluorophore to the defined 30% level (see Methods) was used in the control experiment to reduce the fraction of labeled p53. When p53 oligomers are present, the depolarizing homoFRET in the oligomers should decrease and the fluorescence anisotropy increases [82]. Results with statistical analysis are shown in S4 Fig. Obtained

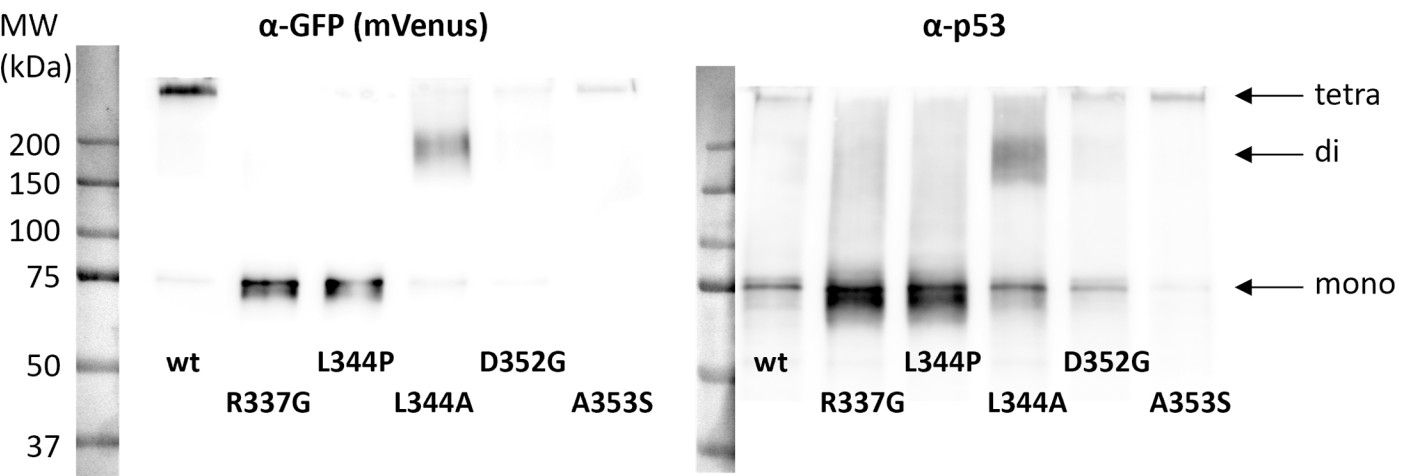

**Fig 3. Immunoblot of GA-crosslinked lysates from HEK-293T cells transfected with mVenus-labeled p53 variants.** Positions of monomers, dimers and tetramers are marked with arrows.

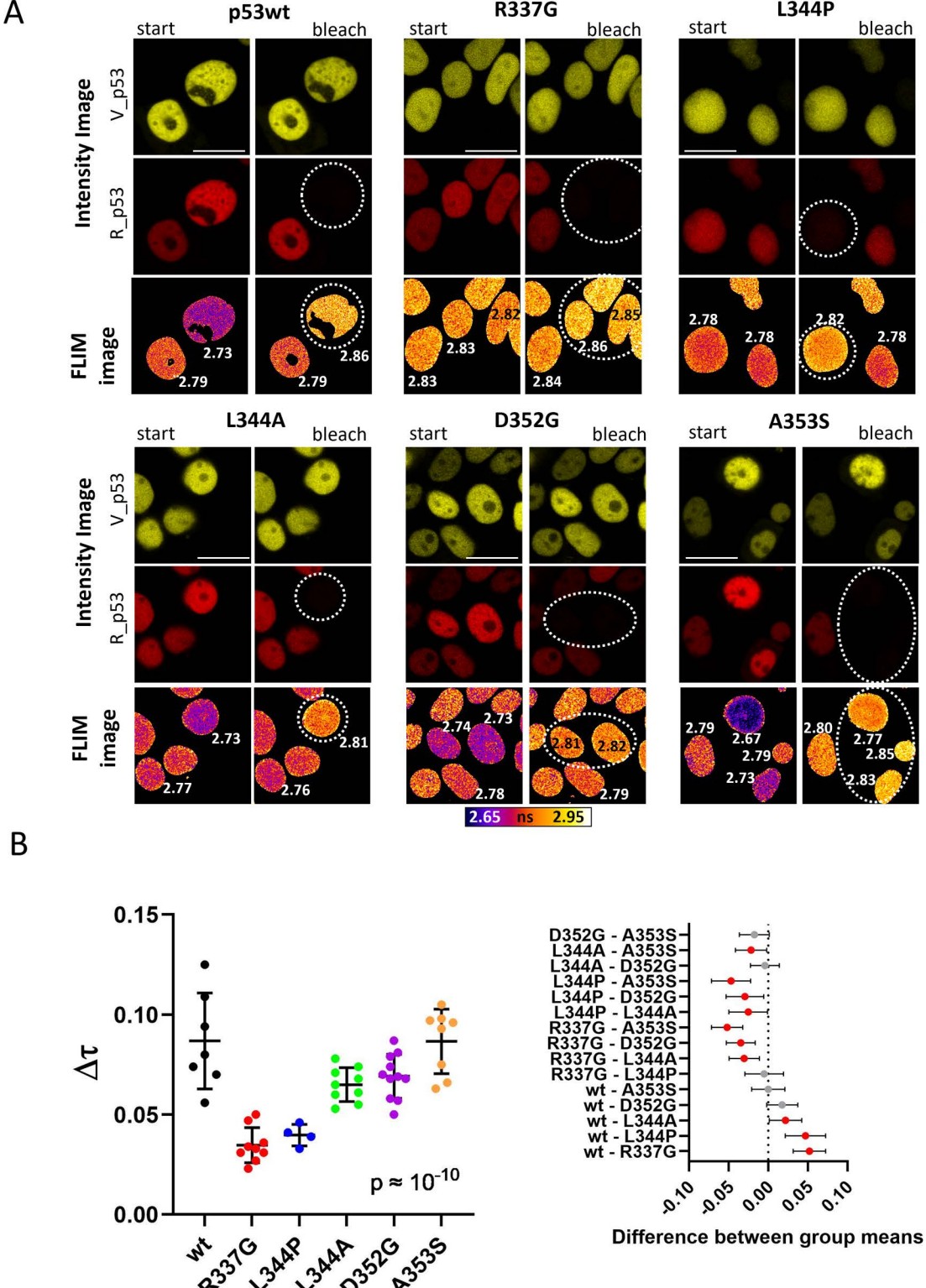

**Fig 4.  A) Intensity (upper two rows of the respective panel) and FLIM (bottom row of the respective panel) images of p53 variants in HEK-293T cells co-transfected with two different color variants of the specific FP-tagged protein (V_p53 donor/R_p53 acceptor).** Left columns (start): initial state, right columns (bleach): images after photodestruction of mRFP1 in cells marked with the dotted line by 561 nm light. The numbers refer to

fluorescence lifetimes in the corresponding cells (in ns). Bar 20 µm. B) Statistical evaluation of the fluorescence lifetime changes in the heteroFRET experiments. Mean values are plotted with ±SD (left), red symbols in the 95% confidence graph (right) mark significant differences between appropriate pairs of samples (p < 0.05).

results agree with the heteroFRET measurements and confirm the monomeric state of the R337G and L344P mutants, since anisotropy changed only slightly after the bleaching. Anisotropies of the L344A, D352G, A353S, and p53wt resembled each other. D352G mutant differed from p53wt insignificantly and intermediate differences in the anisotropy change were observed for L344A and A353S. This indicates higher oligomerization status of these mutants.

Taken together, R337G and L344P were confirmed to be monomeric by multiple methods. From the putative dimers, only L344A was found to be predominantly dimeric. D352G and A353S can likely form higher oligomers.

## Oligomerization of p53 mutants at basal cellular concentrations

The level of FP-labeled exogenous p53 variants in transfected HEK-293T cells is usually higher compared to the physiological conditions. For the FLIM-FRET and anisotropy imaging experiments it is typically necessary to select cells with high enough fluorescence, which means a higher p53 expression. As a consequence, the measured equilibrium may be shifted to higher p53 oligomers compared to physiological conditions. Therefore, it is useful to examine formation of p53 oligomers also at lower p53 concentrations, which can be done by fluorescence cross-correlation spectroscopy (FCCS) [83]. Typical concentrations measurable by FCCS range from nM up to few µM, which are close to the basal cellular p53 concentrations [33,84]. As shown in Fig 5A, oligomerization status of the p53 variants agrees at low concentrations with the results of FRET and crosslinking experiments. Specifically, the dimeric L344A exhibits the cross-correlation amount between the one of the monomeric and multimeric constructs. We did not observe significant difference in the diffusion times ($\tau_D$) of p53wt, D352G and A353S (Fig 5B), which indicates that these oligomers have similar hydrodynamic and interaction properties in nuclei. The measured concentrations of the FP-labeled constructs ranged between 0.1–1.0 µM, as determined form fitted amplitudes of auto-correlation functions, S5 Fig. In summary, the ability of the L344A, D352G, A353S, and p53wt variants to form oligomers in live cells was consistently documented by all experiments in a broad range of concentrations.

## p53 forms oligomers in the cytoplasm

The solution conditions and microenvironment in the cytoplasm dramatically differ from those in the nucleus. p53 concentration is also lower. It is therefore legitimate to search for p53 oligomers in the cytoplasm of cells expressing NPMmut, which facilitates p53 cytoplasmic export. Oligomerization of cytoplasmic p53 in NPMmut-expressing cells was investigated using 3-color labeling of proteins co-transfected into the cell. Two p53wt constructs labeled with a FRET pair (Venus/mRFP1) were directed to the cytoplasm by C_NPMmut (S6 Fig). Interaction between the cytoplasm-localized p53wt molecules was measured by FLIM-FRET. Although the concentration of p53 in the cytoplasm is lower than the one in the nucleus, FRET revealed presence of p53wt oligomers there. Therefore it seems, that oligomerization of p53wt is strong enough to occur regardless of its localization and concentration. Weaker oligomerization was observed for A353S relocalized to the cytoplasm.

## NES significance for cytoplasmic localization of p53

To assess an impact of the individual p53 mutations on the protein localization, we investigated subcellular localization of the FP-labeled mutants in the absence of NPMmut (Fig 6). All mutants localized into the nucleus with slight localization difference between the monomeric and multimeric variants. As seen in Fig 6A, p53wt and oligomeric L344A, D352G, and A353S omit the nucleoli, in contrast to the monomeric R337G and L344P, which exhibit comparable nucleolar and

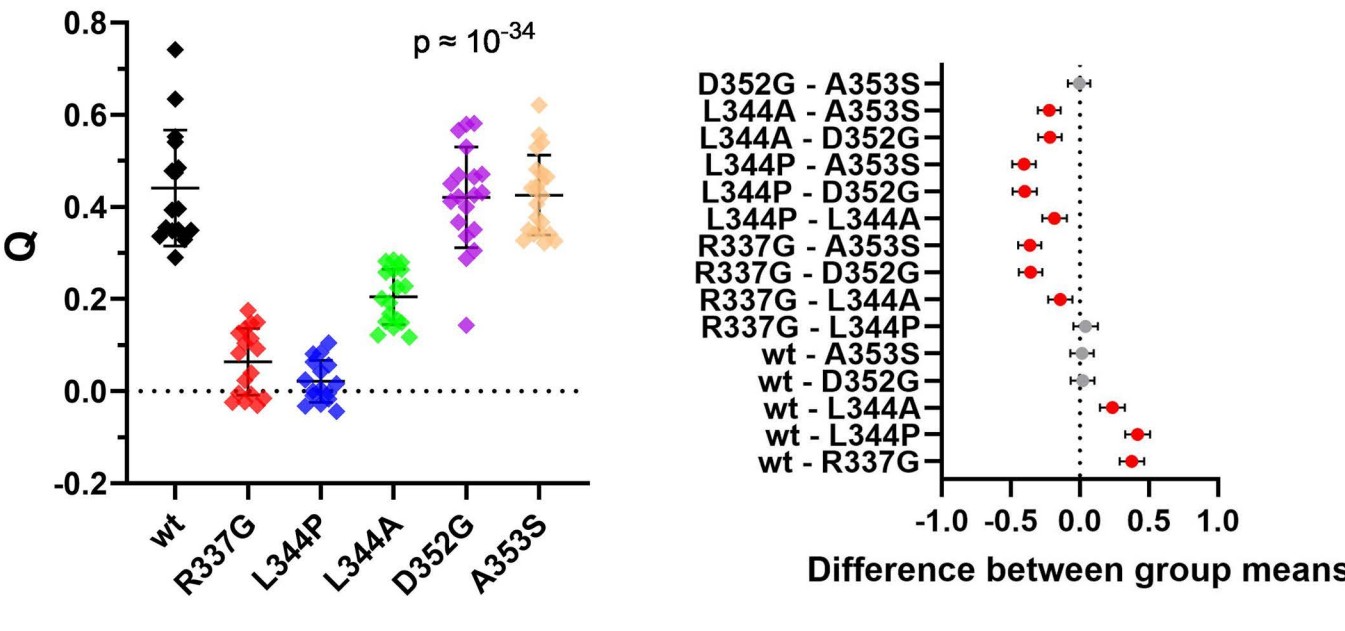

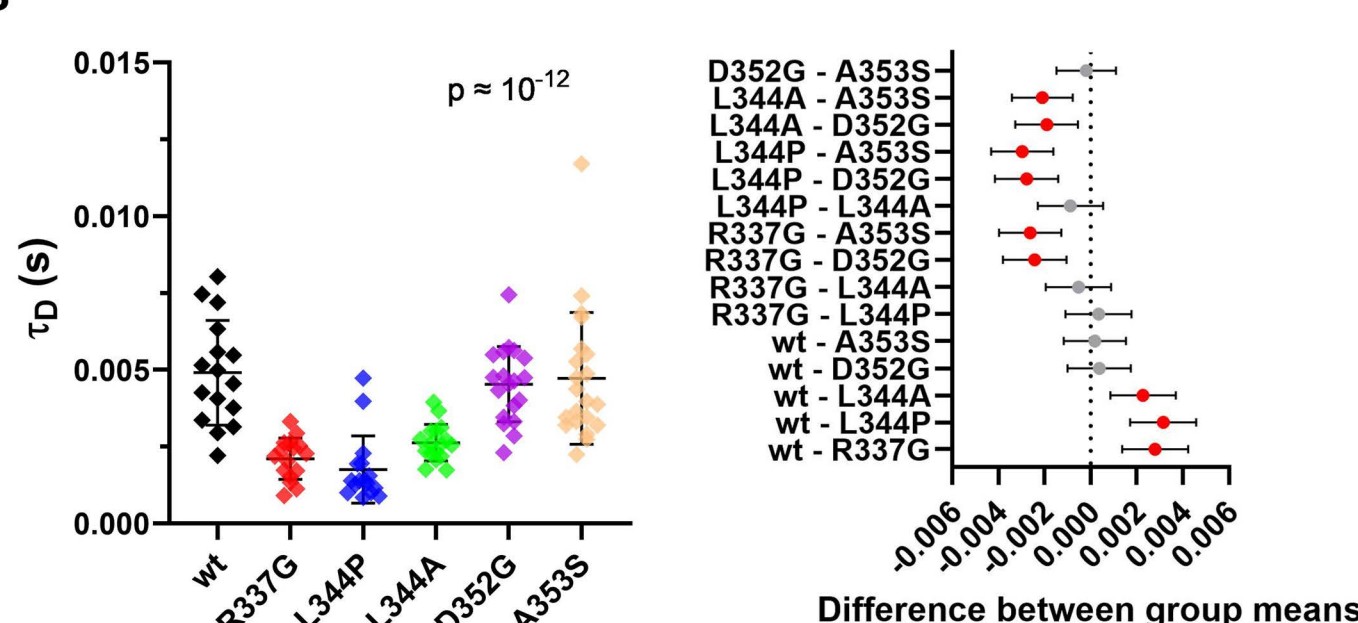

**Fig 5. FCCS measurements in the nucleus of HEK-293T cells co-transfected with a mix of mVenus- and mRFP1-labeled p53 variants.** A) The cross-correlation amount Q was determined for each FCCS measurement. B) The diffusion time $\tau_D$ was calculated from autocorrelation curves recorded in the mVenus detection channel. Mean values are plotted with ±SD (left), red symbols in the 95% confidence graph (right) mark significant differences between appropriate pairs of samples (p < 0.05).

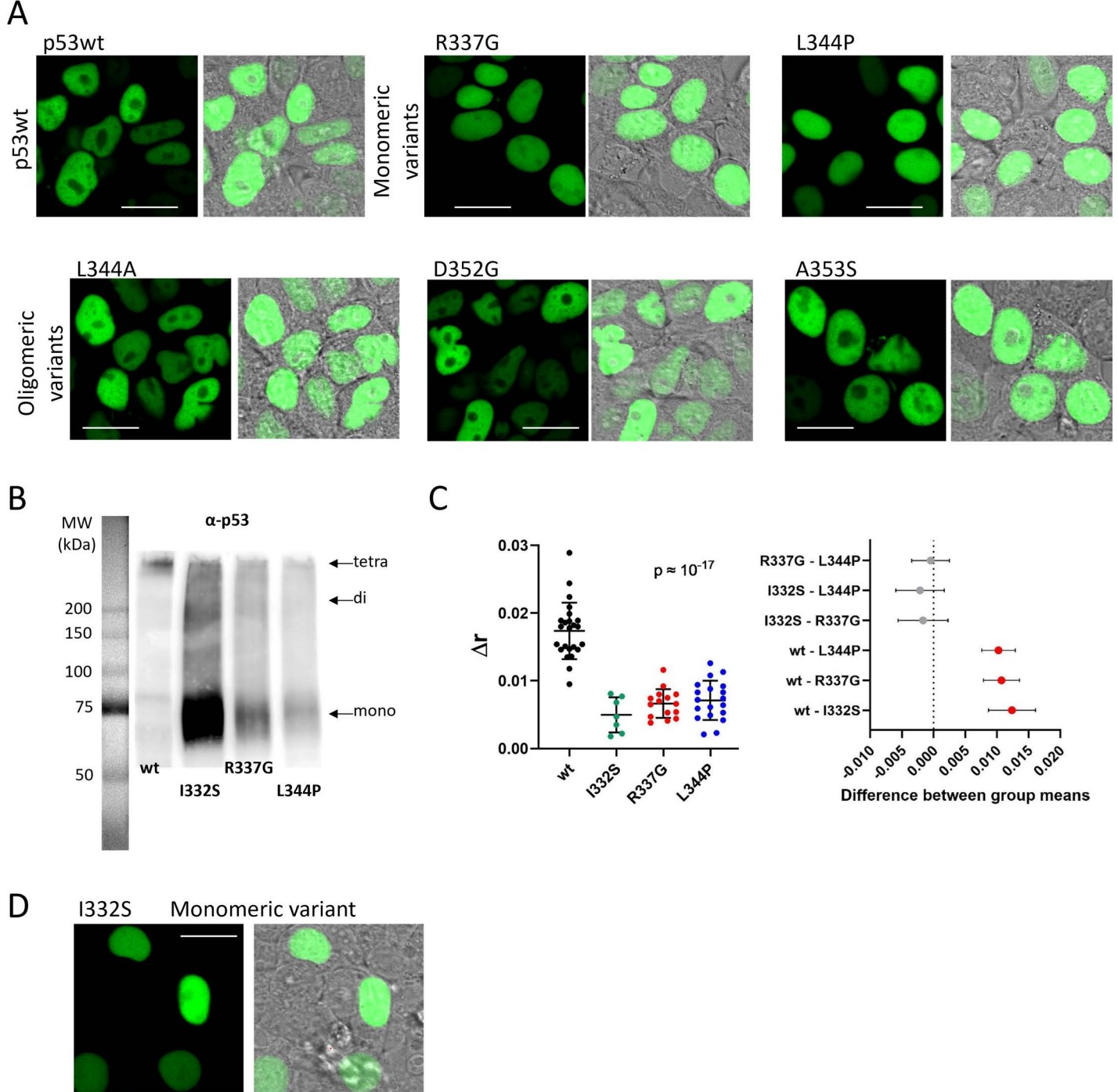

**Fig 6. A) Subcellular localization of p53 mutants tagged with nowGFP in HEK-293T cells.** B) Immunoblot of nowGFP-labeled I332S using GA-crosslinked lysates from HEK-293T cells, Comparison with p53wt and other monomeric variants, C) Comparison of fluorescence anisotropy changes. Mean values are plotted with ±SD, red symbols in the 95% confidence graph (right) mark significant differences between appropriate pairs of samples (p < 0.05)., D) Localization of nowGFP-labeled I332S in HEK-293T cells. Bar represents 20 μm.

nucleoplasmic localization and nearly a homogeneous distribution across the nuclei. Strictly nuclear localization of mono-meric R337G and L344P disagrees with the general picture, that the monomeric p53 exposes its NES that facilitates export to the cytoplasm [44]. In case of monomeric L344P, no conclusion can be made as the main NES is affected by the mutation. We therefore investigated the alternative I332S variant with the intact NES. As seen from Fig 6B and 6C, I332S is monomeric and exhibits the same nuclear localization and distribution as other monomeric R337G and L344P (Fig 6D).

Since p53 carries both NES and NLS, p53 localization must depend on the interplay between these two signals. We therefore mutated the main NLS in the set of p53 variants to check their localization (Fig 7). Oligomeric status of these constructs remained identical to their native NLS-containing counterparts, as documented by immunoblotting of glutaraldehyde-crosslinked samples (Fig 7A).

As expected, we found that all oligomeric NLS-lacking variants became localized solely in the cytoplasm (Fig 7B), which confirms the crucial role of NLS in the nuclear localization of p53. On the other hand, the monomeric NLS-lacking variants were contra-intuitively localized both in the nucleus and the cytoplasm. We can therefore conclude that the NES itself is not strong enough to fully deplete the nucleus from the monomeric p53, which partially localizes to the nucleus even in the absence of NLS.

## Non-transported p53 variants interact weakly with the NPMmut

The p53-NPMmut interaction was tested by co-immunoprecipitation (Fig 8). Consistently with expectations from the localization experiments in Fig 2, attenuation of the interaction between p53 mutants and NPM is clearly visible for all constructs except A353S, which is the only variant colocalizing to the cytoplasm with NPMmut. Although some degree of interaction exists for all constructs, the interaction strength correlates with p53 delocalization and weakly interacting p53 forms are not relocalized. It documents that both the interaction strength and p53 oligomeric form are important for p53 mislocalization.

We have shown previously [62] that the NPMmut-driven p53wt delocalization occurs also in the presence of truncated nonoligomerizing NPMmut (Δ117NPMmut). The fraction of cytoplasmic p53wt was found to be even higher in the presence of Δ117NPMmut compared to the full-length protein, despite the p53-Δ117NPMmut interaction was significantly weaker. We therefore tested the localization and oligomerization of selected p53 variants in the presence of monomeric Δ117NPMmut. It was found that under these conditions even D352G becomes partially localized to the cytoplasm, although to a lesser extent than p53wt or A353S, S7 Fig. Other mutants, the monomeric R337G and L344P, and dimeric L344A, were not pulled-out by non-oligomerizing Δ117NPMmut at all. Similarly to NPMmut, FRET documenting formation of p53 oligomers was detected in the cytoplasm. Altogether, the data suggest that combination of the oligomerization and interaction ability of p53 play a role in its nuclear export in the presence of NPMmut. Oligomerization of NPMmut seems not to be essential for the translocation. On the other hand, oligomerization of p53 is critical, as p53 monomers were never translocated neither with oligomeric nor with monomeric NPM.

## Stability of p53 complexes plays a significant role in the translocation process

We confirmed the predominantly monomeric state of the R337G and L344P variants and the predominantly oligomeric state of p53wt and L344A, A353S and D352G mutants both in live cells and cell lysates. Since our live-cell measurements cannot clearly discriminate between dimers and tetramers, only the L344A mutant can be considered as predominantly dimeric, based on the blotting results. Other mutants seem to assume almost exclusively tetrameric form (Fig 3). However, localization of A353S and D352G in the presence of NPMmut differs considerably.

To explain this difference, we analyzed cell lysates of all monomeric and oligomeric p53 variants by PAGE with various SDS concentrations (Fig 9). Under the native PAGE conditions (0% SDS), surprisingly, no difference between the mutants was found. This indicates that under favorable concentration and solution conditions even apparently monomeric mutants can form complexes, possibly with p53wt and other p53-interacting proteins in the lysate. Increasing SDS concentration

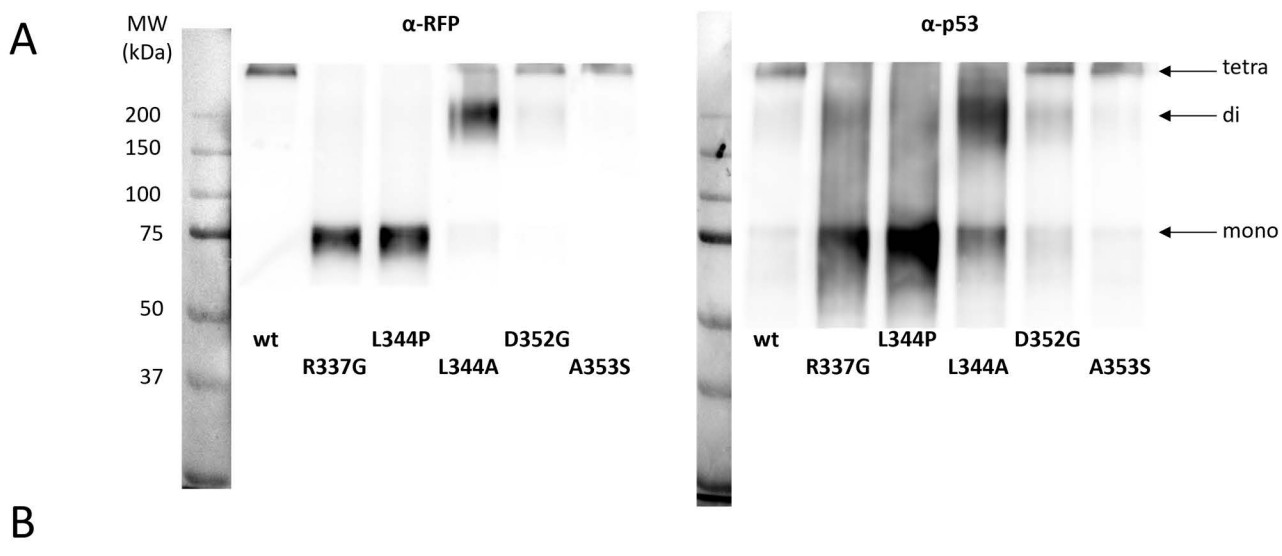

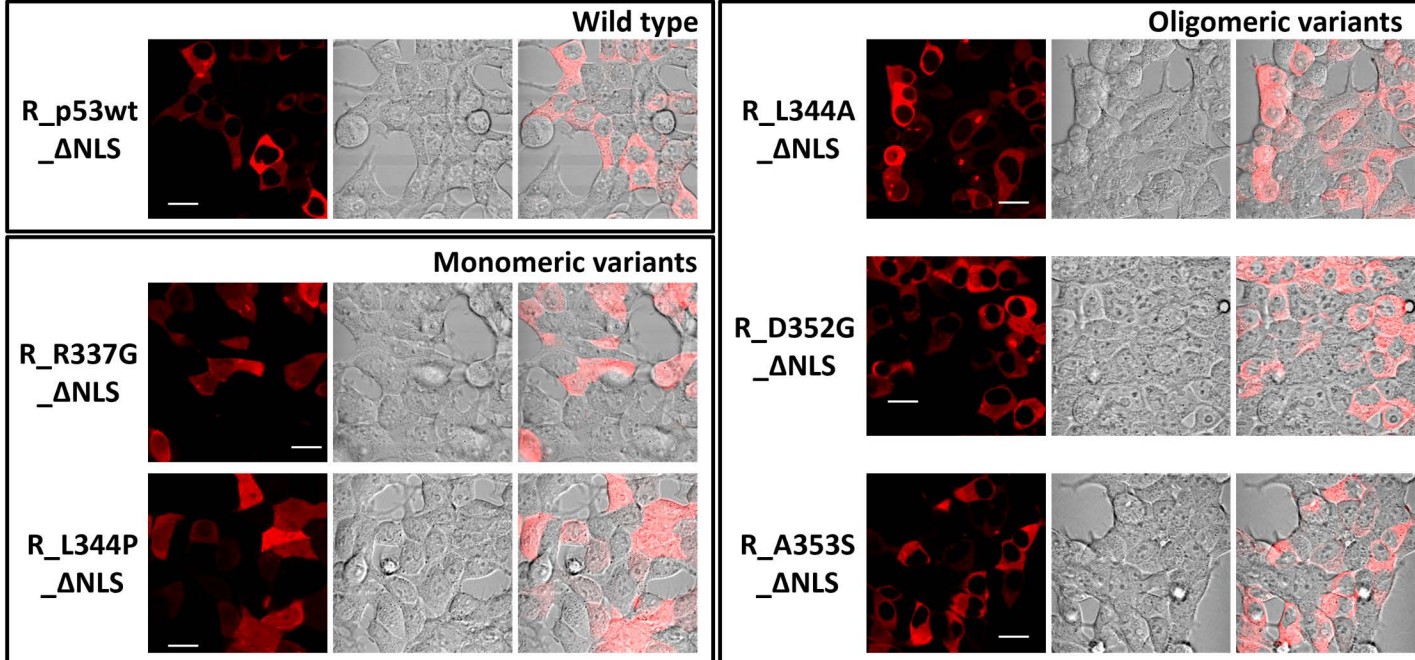

**Fig 7. Characteristics of mRFP1-labeled p53 variants with impaired NLS expressed in HEK-293T cells.** A) Immunoblot of GA-crosslinked lysates with p53 variants detected by anti-RFP (left) or anti-p53 (right) antibody. B) Representative images of the subcellular localization. Bar represents 20 µm.

caused gradual decay of the complexes to monomers. As expected, complexes of the variants classified above as monomeric disintegrated first, already at 0.005% SDS. Oligomeric L344A and D352G revealed similar higher threshold of the complex disintegration. Their complexes dissociated to monomers at 0.01% SDS. Complexes of the remaining two variants, i.e., p53wt and A353S, exhibited the highest stability and decomposed at 0.1% SDS. We can therefore conclude that besides the oligomeric state and the interaction strength with NPMmut also the stability of the p53 complexes plays a significant role in the translocation process.

A

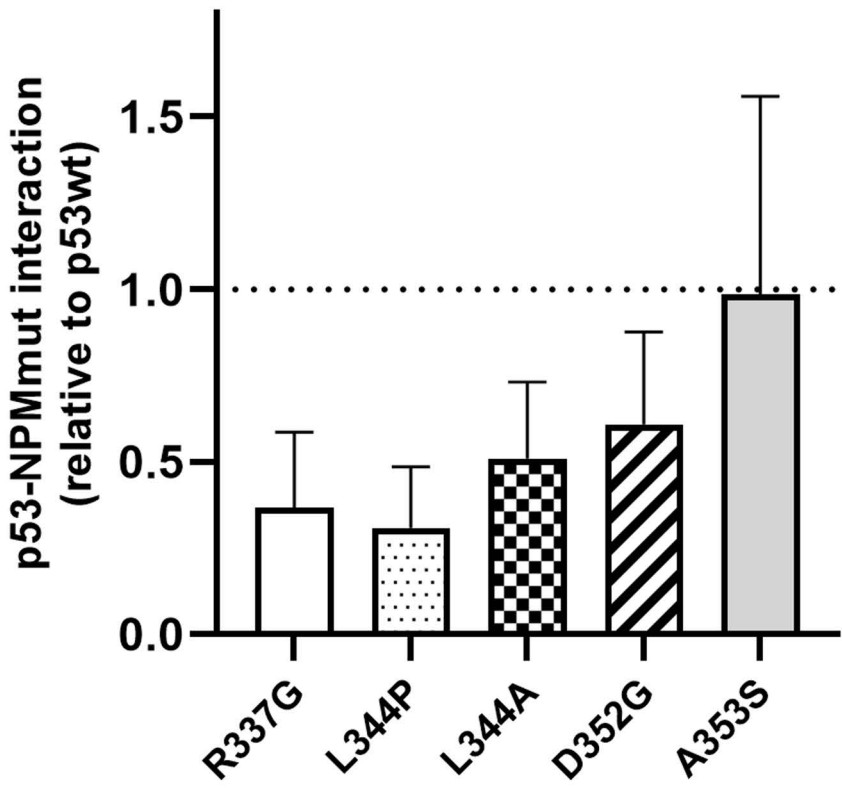

B

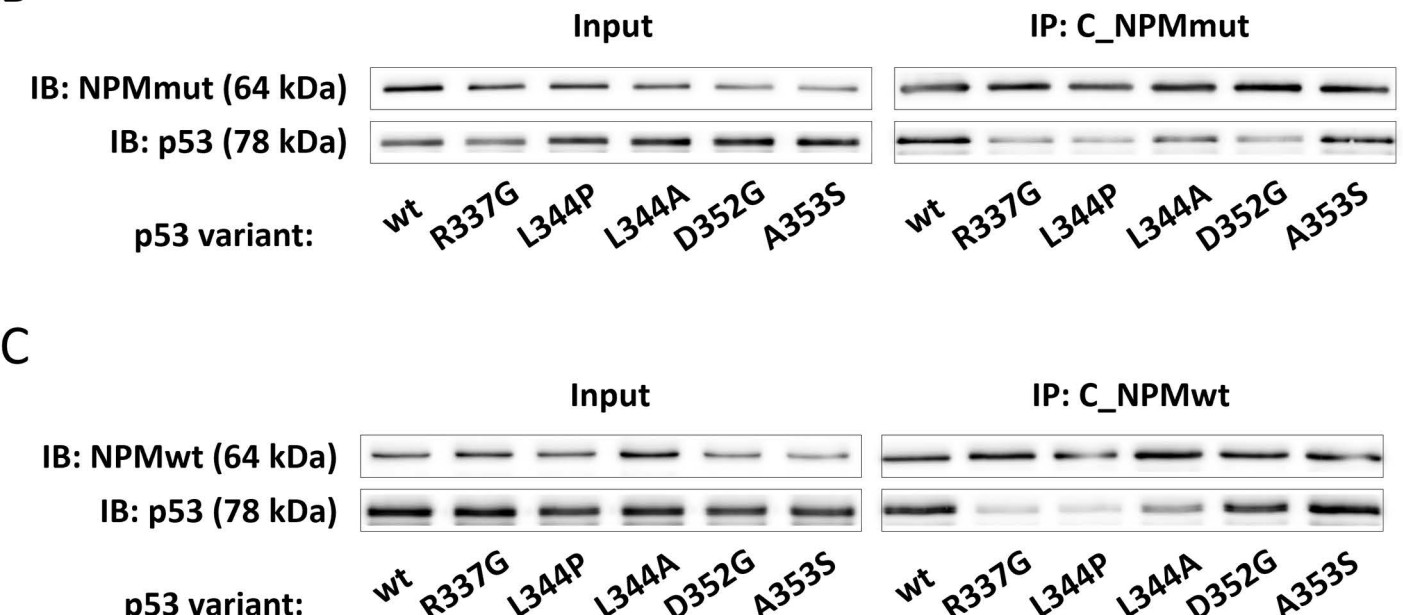

Fig 8. Co-immunoprecipitation of NPM and p53 variants. A) Statistical analysis of p53 immunoblot band intensities co-precipitated with NPMmut relative to the p53wt (mean ± SD from at least 3 independent experiments). B) Representative immunoblot of NPMmut and p53 levels in the total cell lysate

(Input) and NPMmut-precipitate (IP: C_NPMmut). C) Representative immunoblot of NPMwt and p53 levels in the total cell lysate (Input) and NPMwt-precipitate (IP: C_NPMwt).

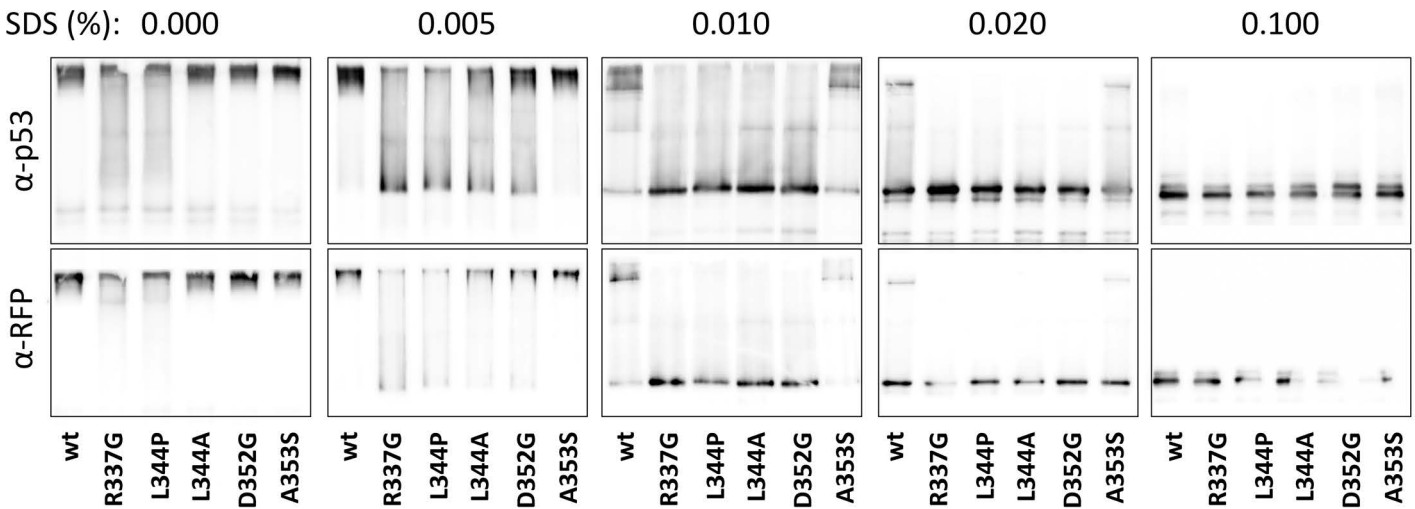

**Fig 9. Stability analysis of mRFP1_p53 variants.** Transfected HEK-293T cells were lysed under native conditions and seminative PAGE with varying SDS concentration was performed. Detection of p53 (upper row) and RFP (lower row) revealed different SDS thresholds for decomposition of p53 complexes.

## Charge-swapping mutation R337D-D352R lowers stability of p53 oligomers

As predicted by MD simulations, the salt bridge between R337 and D352 should be essential for the stabilization of the p53 dimer [85]. Indeed, a strong destabilization effect of both R337 and D352 mutations was documented by native MS, which was attributed mainly to disruption of the stabilizing salt bridge between R337 and D352 in the p53wt monomer that supports the dimer and tetramer structure [86]. Since mutation to Gly in both residues affected stability of the p53 oligomers and the ability to be pulled out to the cytoplasm by NPMmut, we tested whether charge swapping between R337 and D352 by the double mutation R337D-D352R could rescue the p53wt phenotype (S8 Fig). The mutation was expected to restore the stabilizing R337-D352 salt bridge and both p53 oligomerization and its cytoplasmic localization in the presence of NPMmut. We therefore prepared R337D and D352R single mutants and verified their oligomerization. As seen from S8 Fig, R337D is monomeric, which agrees with the R337G characteristics and confirms an important role of R337 residue, e.g., in the Li-Fraumeni syndrome. D352R forms multimers similar to D352G, as verified by fluorescence anisotropy (S8 Fig (A)). Also the nuclear localization of R337D and D352R corresponds with the one of R337G and D352G (S8 Fig (B)) and seminative PAGE shows similar destabilization of D352R and D352G compared to variants with the wt phenotype (S8 Fig (D)).

Although the salt bridge restoration should rescue the wt phenotype, we have not observed the expected cytoplasmic localization of the double mutant in the presence of NPMmut (S8 Fig (B)). We obtained the double mutant both via the single-mutated R337D and D352G variants. Both constructs provided the same phenotype and remained in the nucleus in the presence of NPMmut. The observation is consistent with results of emission anisotropy (S8 Fig (A)), GA crosslinking (S8 Fig (C)), and seminative SDS-PAGE (S8 Fig (D)), which document mainly monomeric nature of the double mutant.

## Discussion

The molecular mechanisms regulating the transcriptional activity and degradation of p53 are extremely complex and p53 oligomerization plays a crucial role in many of them [28,87]. Among others, defective oligomerization is the basis of Li-Fraumeni syndrome [41]. We have previously documented aberrant cytoplasmic localization of p53 in the presence of NPMmut [62]. Here we aimed to elucidate the role of p53 oligomerization in its cytoplasmic mislocalization induced by the NPM mutation. We prepared a set of p53 constructs with point mutations in the Tet domain that are expected to affect specifically the oligomerization. In GA crosslinking (Figs 3 and S3), we have shown that oligomerization of some of these p53 variants disagree with the published results. Discrepancy was found for the D352G and A353S constructs, which were reported to be dimeric [81], whereas we found their arrangement in higher oligomers. The difference could stem from the system that was developed to test oligomerization of the p53 variants. While Kawaguchi et al. [81] used yeast, we used human cells. Moreover, verification of the whole *TP53* gene sequence for high number of constructs is difficult to achieve in the yeast high-throughput system. In our site-directed mutagenesis we found considerable amount of accidental amended mutations, even in domains far from the Tet domain that could generally bias the oligomerization outcome.

We thoroughly tested all constructs for oligomerization in live cells. Both FLIM-FRET and anisotropy measurements clearly distinguished between monomer- and oligomer-forming mutants (Figs 4 and S4). Combination of the imaging and crosslinking results therefore confirmed both *in vivo* monomeric state of R337G and L344P and predicted dimeric state of L344A in transfected cells. *In vivo* oligomerization characteristics of D352G and A353S were found between those of oligomeric p53wt and dimeric L344A, which agrees with the crosslinking experiments.

As the oligomerization depends on the concentration of the interacting molecules, we verified results by point FCCS measurements in live cells with significantly lower exogenous p53 concentrations (Figs 5 and S5). We are aware that a potential interaction of unlabeled endogenous p53wt with the exogenous constructs could partially bias the cross-correlation amount to lower values by formation of mixed complexes with labeled p53 and the diffusion times $T_D$ to lower values due to a molecular weight of the missing fluorescent tag on endogenous p53 monomers. Nevertheless, the FCCS results are consistent with outcomes of all other methods and document that our set of mutants comprises two monomers, one dimer and two constructs with oligomerization properties similar to p53wt.

The localization analysis indicates that besides p53wt, only the A353S mutant with oligomerization similar to the one of wt was pulled-out to the cytoplasm in the presence of NPMmut (Fig 2). On the contrary, D352G, which displayed almost identical oligomerization as A353S, was not co-transported. It seems therefore, that the high oligomeric state of p53 is a necessary but not sufficient condition for its mislocalization in the context of this pathogenic mutation. The difference between D352G and A353S was clearly seen on seminative SDS-PAGE (Fig 9) where oligomers of A353S exhibited the highest wt-like stability at increased concentration of denaturing SDS. While complexes of all other mutants dissociated below 0.01% SDS, oligomers of A353S and p53wt were stable up to 0.1% SDS. The sufficient stability of the p53 oligomer therefore seems to be another key requirement for cytoplasmic translocation by NPMmut. Although the majority of the examined p53 mutants interact with NPMmut, only A353S exhibiting the strongest association to NPMmut is relocalized (Fig 8). The interaction strength and the stability of the p53 oligomers correlate and might be interlinked. Both are important for p53 mislocalization.

We have previously shown that p53wt can be translocated to the cytoplasm also by monomeric Δ117NPMmut, which lacks the oligomerization domain [62]. Here, using Δ117NPMmut, we achieved partial cytoplasmic localization of not only p53wt and A353S but also D352G (S7 Fig). It seems therefore that the oligomeric status of NPMmut is not critical for the relocalization and the interaction with p53 occurs on the interface of the NPMmut monomer. Moreover, we found that translocated variants oligomerize in the cytoplasm, even at very low concentration there.

The interplay between NES and NLS signaling should modulate the resulting p53 localization in the cell. Stommel et al [44] proposed a model of NES masking, where monomeric p53 exposing the unmasked NES should be readily present in the cytoplasm. Also masking of NLS was considered in Liang & Clarke [88], who indicate that p53 monomers can enter the

nucleus easier. We observed exclusively nuclear localization of all monomeric constructs independently of the NES integrity (Fig 6). We therefore hypothesize that NES functions mainly in cooperation with NLS to ensure dynamic subcellular distribution of p53 and its relation to p53 oligomeric state is still unclear. Additional mutation affecting the NLS domain, which we introduced to our constructs, led to exclusively cytoplasmic localization of all oligomeric forms, which also disagrees with the hypothesis of the NES masking in oligomers. The NLS-lacking monomeric mutants were distributed both in the nuclei and cytoplasm, independently of the NES impairment (Fig 7). The cellular localization of FP-labeled p53 monomers (~70 kDa) could explain their free diffusion through the nuclear pore complex with the cutoff size of 110 kDa [89], which short-circuits the nuclear transport. Neither labeled dimers nor higher oligomers can pass the nucleopore complex.

A functional NLS is thus extremely important for the nuclear localization of p53 oligomers, while the nuclear localization of monomers seems to be independent of the main NLS. The monomeric p53 variants with intact NLS exhibited a specific localization pattern in the cell nucleus with homogenous distribution in all nuclear regions including the nucleolus (Fig 6 (A, D)). Although p53wt resides mostly in the nucleoplasm, its nucleolar localization was previously reported in some special cases [90–92], including poly-Ubq p53[54]. Nevertheless, poly-ubiquitinylation might not be the only reason for the nucleolar localization since ubiquitinylation defect was reported for some nucleolar *Tet*-domain mutants [87]. Detailed localization mechanism of the monomeric p53 variants remains unclear, however their higher mobility together with specific nucleolar interactions are likely important.

Residues R337 and D352 were reported to play a key role in p53 conformation due to the formation of a salt bridge between them [85,86]. Mutation of any of these residues should break the bridge with negative impact on p53 oligomerization and the complex stability. Indeed, we confirmed such behavior (Fig 9). Rigoli et al. [31] found a partial rescue of the transactivation ability of the R337D-D352R double-mutant, which was attributed to the restoration of the salt bridge after the amino acids exchange. Based on their findings, we strived to see also a rescue of oligomerization and localization of this double-mutant (S8 Fig). Similarly to R337G, R337D was also found to form almost exclusively monomers and both variants localized strictly in the nucleus even in the presence of NPMmut. D352R behaved similarly as D352G, i.e., it exhibited the same oligomerization and localization properties. Nevertheless, the charge swapping in the R337D-D352R double mutant resulted in the monomeric phenotype, the same as R337 mutants. The amino acid exchange at positions 337 and 352 therefore does not restore the wild-type behavior in terms of oligomerization and localization and a functional restoration of the salt bridge in this mutant is thus disputable. Results confirmed the prominent role of the R337 residue for the p53 structure and function.

To correlate pathological consequences of our p53 variants with their oligomerization and localization characteristics, we performed search in the AlphaFold2 Protein Structure database [93]. Based on a protein structure, this AI-based system predicts mutation pathogenicity score related to severity of specific point mutations. Table 1 shows that perturbed p53 oligomerization is correlated with high pathogenicity. We hypothesize that in AML, potential synergy between cellular

**Table 1. Pathogenicity score of different p53 point mutations according to AlphaFold2 [93].**

| p53 mutation | Pathogenicity score by AlphaFold* | Oligomeric state | Cytoplasmic localization with NPMmut |
|---|---|---|---|
| I332S | 0.997 | monomer | – |
| R337G | 0.950 | monomer | – |
| R337D | 0.975 | monomer | – |
| L344P | 0.990 | monomer | – |
| L344A | 0.925 | dimer | – |
| D352G | 0.448 | oligomer | – |
| D352R | 0.831 | oligomer | – |
| A353S | 0.009 | oligomer | + |

*Score 0-0.340 = likely benign, 0.340-0.564 = uncertain, > 0.564 = likely pathogenic

mislocalization of p53 by NPMmut and p53 abnormality might dramatically enhance severity of less pathogenic, or seemingly benign, p53 forms with closely wt-like behavior.

## Conclusion

Relation between p53 oligomerization and function has been widely documented. Here we showed that p53 oligomerization is essential but not sufficient for its cytoplasmic co-transport with AML-specific NPM mutation. Stability of p53 oligomers and cooperation of localization and export signals are other important factors playing a role in this process. This observation uncovers alternative therapeutic targets for p53 nuclear retention in AML cells with NPM mutation.

## Supporting information

**S1 Table. A) List of constructs.** B) Primers used for construction of plasmids.
(DOCX)

**S2 Table**  p-values from the Tukey-Kramer post-analysis for graphs in Figs 4, 5, 6, S4 and S8
(DOCX)

**S1 Fig. Schematic representation of NPM1 protein domains with indicated positions of the NES and NLS.** The alternative NES formed by the AML-related mutation is included in the scheme.
(TIF)

**S2 Fig. Cellular localization of mRFP1-labeled variants of p53 (R_p53) co-expressed with Cerulean labeled NPMwt (C_NPMwt).** A) p53wt, B) monomeric variants, and C) putative dimeric variants. Bar represents 20 μm. (D) Expression of mRFP1-labeled NPM variants (wt or mut) in transfected HEK-293T cells.
(TIF)

**S3 Fig. Immunoblot of p53 variants using GA-crosslinked lysates from HEK-293T cells transfected with nowGFP-labeled p53 variants.**
(TIF)

**S4 Fig. Steady-state anisotropy imaging.** A) Demonstration of the homoFRET effect on nuclei of HEK-293T cells. The presence of the energy transfer manifests itself as an anisotropy increase in the photobleached sample, bar represents 10 μm. B) Representative anisotropy measurements in nowGFP-labeled p53 variants. Dotted line marks bleached cells with emission reduced to the 30% of the initial intensity. Bar represents 20 μm. C) Statistical evaluation of the anisotropy experiments (2–7 cells per image in 2–5 independent measurements). Mean values are plotted with ±SD (left), red symbols in the 95% confidence graph (right) mark significant differences between the appropriate pairs of samples ($p < 0.05$).
(TIF)

**S5 Fig. Typical autocorrelation and cross-correlation curves measured in nuclei of HEK-293T cells co-transfected with a mix of mVenus- and mRFP1-labeled p53 variants.** Solid lines are data fits.
(TIF)

**S6 Fig. Oligomerization of p53wt and A353S pulled-out to the cytoplasm by NPMmut.** A) Intensity images of C_NPMmut, V_p53 and R_p53 before (START) and after (BLEACH) the photodestruction of mRFP1 by intense 561 nm irradiation. B) FLIM image before (left column) and after (right column) photodestruction of mRFP1 in cells marked with the dotted line. Numbers indicate lifetime in the cytoplasm. Bar represents 20 μm.
(TIF)

**S7 Fig. Representative localization and FRET-FLIM images of selected p53 variants in the presence of mono-meric Δ117NPMmut.** A) Subcellular localization of C_Δ117NPMmut, V_p53wt, and R_p53wt (1st, 2nd and 3rd column, respectively) before (START) and after (BLEACH) photodestruction of mRFP1 acceptor by 561 nm light. B) FLIM image before (START) and after (BLEACH) the mRFP1 photobleaching in cells bordered by the dashed line. Numbers refer to the emission lifetime in the cytoplasm. Bar represents 20 μm.
(TIF)

**S8 Fig. Characteristics of nowGFP-labeled R337D, D352R and double mutants R337D-D352R compared to wt: anisotropy (A), localization in the presence of C_NPMmut, bar represents 20μm (B), p53-immunoblot of GA-crosslinked lysates (C) and of native lysates on gel with reduced (0.01%) SDS concentration (D).** The difference between R337D-D352R and D352R-R337D variants is in the starting p53 variant that was used for the introduction of the second mutation. In the anisotropy graph (A, top), mean values are plotted with ±SD. Red symbols in the 95% confidence graph (A, bottom) mark significant differences between appropriate pairs of samples (p < 0.05).
(TIF)

## Acknowledgments

Authors thank Marie Olšinová from Imaging Methods Core Facility at BIOCEV for her support with the FCCS data acquisition, and Pavla Pecherková from Institute of Hematology and Blood Transfusion for help with the statistical analyses.

## Author contributions

**Conceptualization:** Ales Holoubek, Dita Strachotova, Barbora Brodská, Petr Herman.

**Data curation:** Ales Holoubek, Dita Strachotova, Katerina Wolfova, Petra Otevrelova, Sara Belejova, Pavla Roselova, Ales Benda, Barbora Brodská, Petr Herman.

**Formal analysis:** Ales Holoubek, Ales Benda, Barbora Brodská, Petr Herman.

**Funding acquisition:** Barbora Brodská, Petr Herman.

**Investigation:** Ales Holoubek, Dita Strachotova, Katerina Wolfova, Petra Otevrelova, Sara Belejova, Barbora Brodská, Petr Herman.

**Methodology:** Ales Holoubek, Dita Strachotova, Petra Otevrelova, Pavla Roselova, Ales Benda, Barbora Brodská, Petr Herman.

**Project administration:** Barbora Brodská, Petr Herman.

**Resources:** Petr Herman.

**Supervision:** Barbora Brodská, Petr Herman.

**Writing – original draft:** Ales Holoubek, Dita Strachotova, Barbora Brodská, Petr Herman.

**Writing – review & editing:** Ales Holoubek, Dita Strachotova, Katerina Wolfova, Petra Otevrelova, Sara Belejova, Pavla Roselova, Ales Benda, Barbora Brodská, Petr Herman.

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
