## [Decision Letter · Decision Letter 0]

12 Jan 2025

PONE-D-24-49808Correlation of p53 oligomeric status and its subcellular localization in the presence of the AML-associated NPM mutantPLOS ONE

Dear Dr. Brodská,

Thank you for submitting your manuscript to PLOS ONE. After careful consideration, the reviewers feel that although the article has interesting findings but does not fully meet PLOS ONE’s publication criteria as it currently stands. Therefore, we invite you to submit a revised version of the manuscript that addresses the points raised during the review process.

We look forward to receiving your revised manuscript.

Kind regards,

Swati Palit Deb

Academic Editor

PLOS ONE

**Journal Requirements:**

The work was supported by the Czech Science Foundation – grant No 22-03875S (PH). BB was supported by MH CZ – DRO (IHBT, 00023736), the project for conceptual development of the research organization. 

Authors thank Marie Olšinová from Imaging Methods Core Facility at BIOCEV, institution supported by the MEYS CR (LM2023050 Czech-BioImaging), for her support with FCCS data acquisition.

The work was supported by the Czech Science Foundation – grant No 22-03875S (PH). BB was supported by MH CZ – DRO (IHBT, 00023736), the project for conceptual development of the research organization. 

Reviewers' comments:

Reviewer's Responses to Questions

**Comments to the Author**

1. Is the manuscript technically sound, and do the data support the conclusions?

Reviewer #1: Yes

Reviewer #2: Yes

2. Has the statistical analysis been performed appropriately and rigorously? 

Reviewer #1: Yes

Reviewer #2: No

3. Have the authors made all data underlying the findings in their manuscript fully available?

Reviewer #1: Yes

Reviewer #2: Yes

4. Is the manuscript presented in an intelligible fashion and written in standard English?

Reviewer #1: Yes

Reviewer #2: Yes

5. Review Comments to the Author

**Reviewer #1: ** The work represents a thorough analysis of the extend to which different p53 mutant, in particular those involved in oligomerisation are affected by the leukaemic NPCmut mutant.

The work is solid and well analysed but could do with some additional information as per my comments below.

1. What is the impact of the mutants - especially the set mutant on ability to interact/oligomerise with p73 or p63?

2. the oligmeric status of NPM is not well described - a similar diagram for NPM domain structure would be helpful and a little more introduction and discussion on this.

3. The authors state their own previous evidence on NPMmut being monomeric and cytoplasmic but NPM oligmerisation is regulated by AKT phosphorylation - this should be discussed. This would also allow them to extrapolate to other cancers where MAPK/AKT signalling is active and the impact of their work more broadly applicable across cancer where p53 mutants are found.

**Reviewer #2:**  In this manuscript Holoubek et al initiated a study to determine the ability of mutated p53 to oligomerize and translocate into the nucleus and or cytoplasm. Mutations tested include monomeric R337G and L344P, dimeric L344A, and multimeric D352G and A353S mutations. These mutations were tested in the context of a NPMmut , which is a mutant of NPM1 that is found in AML and results in the cytoplasmic localization of p53 (through the NPM-p53 interaction) rendering it a LOF. These studies focus on try to understand how NPMmut could pull out p53. This study is important and therefore publishable provided a few issues are resolved.

Major issues

Figure 2 is not properly controlled. In order to determine the effects of NPMmut on p53 localization a wt NPM would need to be also tested. Western blot is needed to ensure that the 2 cell lines expressing NPMmut and NPM wt are at the same expression levels.

Missing from the study are in vitro interaction studies between the mutant p53 proteins and NPMmut, - as well as the NPM WT protein. Studies in Figure 8 use NPMmut but not the wt NPM. Is there a correlation between defects in localization to cytoplasm and the NPMmut (but not wt NPM) and p53 mutation status?

Ttests are used throughout. Tables are shown which describe the results of many test per experiment. This is an incorrect stastical analysis. AMOVA analysis need to be done. A stastician needs to be consulted for the analysis of these figures and added to the authors of the paper.

Minor Issues

FILM is not defined in the abstract

Figure 1 is confusing. The magnified region is ambiguous regarding its position relative to the full length protein. Is the magnified region within the hash marks? Or is it starting at the end of the grey portion, and ending with the C-terminal domain. Can the amino acid numbers be added to the beginning and end of the cutout region.

6. PLOS authors have the option to publish the peer review history of their article (what does this mean? ). If published, this will include your full peer review and any attached files.

**Do you want your identity to be public for this peer review?** For information about this choice, including consent withdrawal, please see our Privacy Policy .

Reviewer #1: No

Reviewer #2: No

---

## [Author Response · Author response to Decision Letter 1]

31 Jan 2025

Reviewer #1: The work represents a thorough analysis of the extend to which different p53 mutant, in particular those involved in oligomerisation are affected by the leukaemic NPCmut mutant.

The work is solid and well analysed but could do with some additional information as per my comments below.

1. What is the impact of the mutants - especially the set mutant on ability to interact/oligomerise with p73 or p63?

Thank you for this important and inspiring point. Unfortunately, in order to keep this work reasonably concise and well focused, we have not tested additional proteins from a family of p53 homologs. This suggested important aspect might be a subject of another separate study.

2. the oligmeric status of NPM is not well described - a similar diagram for NPM domain structure would be helpful and a little more introduction and discussion on this.

Diagrams of the NPM structure can be found in many publications, e.g. in Holoubek et al 2021 or Arregi et al 2015. We included such a diagram, based on Sasinkova 2022, in Fig. S1. In addition, essential NPM domains are specified on p.8 of the revised manuscript.

3. The authors state their own previous evidence on NPMmut being monomeric and cytoplasmic but NPM oligmerisation is regulated by AKT phosphorylation - this should be discussed. This would also allow them to extrapolate to other cancers where MAPK/AKT signalling is active and the impact of their work more broadly applicable across cancer where p53 mutants are found.

Thank you for this remark, which could also open a new research direction. Although interesting, one should be very cautious to generalize and extrapolate our AML-specific findings for other cancers, as the NPMmut occurs almost exclusively in AML. Nevertheless, we never stated that NPMmut is monomeric in the cytoplasm. The opposite is true. We have previously proved that NPMmut oligomerizes in the cytoplasm (Sasinkova 2021). This is now explicitly stated on the p.8 of the revised manuscript. Only the truncated NPMmut molecule marked here as Δ117NPMmut exhibits monomeric behaviour as described in (Sasinkova 2021). We have shown in Fig. S7 that the mislocalization pattern of our p53 variants in the presence of this monomeric NPM mutant is not significantly affected. We therefore concluded that it seems that the oligomeric status of NPMmut is not critical for the relocalization of p53 variants (Discussion, p.30 of the marked manuscript.). We also slightly reworded the abstract to avoid a potential confusion.

Specific inhibition of NPM oligomerization by AKT could be an interesting subject of another study. AKT was reported to inhibit NPM oligomerization due to its phosphorylation at Ser48 (Mitrea 2014), which had further consequence in p53wt stabilization and tumor suppression, and also in accumulation of specific p53 mutant (in-frame deletion of Thr126, Hamilton 2014). It confirms the important role of NPM oligomerization in tumorigenesis. However, several other NPM phosphorylation sites (e.g. Ser4, Thr95, Ser125 or Thr199) are regulated by other kinases, in particular by CKII and cdk1, and these modifications are also essential for the cell proliferation and apoptosis. Similarly, as can be seen from our set of p53 mutants, individual p53 mutations can manifest themselves differently.

Reviewer #2: In this manuscript Holoubek et al initiated a study to determine the ability of mutated p53 to oligomerize and translocate into the nucleus and or cytoplasm. Mutations tested include monomeric R337G and L344P, dimeric L344A, and multimeric D352G and A353S mutations. These mutations were tested in the context of a NPMmut , which is a mutant of NPM1 that is found in AML and results in the cytoplasmic localization of p53 (through the NPM-p53 interaction) rendering it a LOF. These studies focus on try to understand how NPMmut could pull out p53. This study is important and therefore publishable provided a few issues are resolved.

Major issues

Figure 2 is not properly controlled. In order to determine the effects of NPMmut on p53 localization a wt NPM would need to be also tested.

NPMwt is always present in AML cells expressing NPMmut (Falini et al 2007). The same is true for cells transfected with exogenous NPMmut. We have previously shown mainly nuclear p53wt localization in the presence of endogenous and exogenous NPMwt (Holoubek et al 2021), which contrasts with the cytoplasmic localization of p53wt in the presence of NPMmut. For convenience of readers, we have included in the supplemental material the localization figure of p53 variants in the presence of NPMwt (Figure S2).

Western blot is needed to ensure that the 2 cell lines expressing NPMmut and NPM wt are at the same expression levels.

The expression levels of NPMwt and NPMmut are comparable, as shown e.g. in (Brodska et al 2017, Sasinkova et al 2018, Holoubek et al 2021). Nonetheless, we included WB documenting comparable expression levels of the NPM variants in the supplemental Figure S2.

Missing from the study are in vitro interaction studies between the mutant p53 proteins and NPMmut, - as well as the NPM WT protein. Studies in Figure 8 use NPMmut but not the wt NPM. Is there a correlation between defects in localization to cytoplasm and the NPMmut (but not wt NPM) and p53 mutation status?

This paper is focused mainly on the mislocalization of p53 variants in the presence of NPMmut. As shown in the new Fig. S2, this mislocalization never happens in the presence of NPMwt for none of the p53 variants. To illustrate identical affinity of p53 variants to both NPMwt and NPMmut, we included representative WB image from immunoprecipitation of samples cotransfected with NPMwt and p53 variants to the Fig. 8. We are aware that the interaction of p53 with NPMwt could under physiological conditions shift, to some extent, the equilibrium and potentially facilitate partial nucleolar retention of p53. Nevertheless, in the transfected cells used in this manuscript the relative level of endogenous NPMwt is too low to bias the reported results and conclusions. The same holds for a small fraction of the mixed NPMmut/NPMwt oligomers in the cytoplasm. Those are the reasons why the interaction strength between p53 variants and NPMwt was not studied in detail in this paper.

Ttests are used throughout. Tables are shown which describe the results of many test per experiment. This is an incorrect stastical analysis. AMOVA analysis need to be done. A stastician needs to be consulted for the analysis of these figures and added to the authors of the paper.

According to the reviewer´s requirement, the ANOVA analysis with multiple comparison post-analysis was done in the revised manuscript. Overall ANOVA p-values were added to the original graphs and tables with individual p-values were replaced by 95% confidence value graphs. Numeric p-values comparing individual sample pairs were shifted to the supplement as Table S2. The description of the statistical analysis was updated in the Material and Methods section and noted in figure legends.

Minor Issues

FILM is not defined in the abstract

The meaning of the FLIM abbreviation was added to the abstract.

Figure 1 is confusing. The magnified region is ambiguous regarding its position relative to the full length protein. Is the magnified region within the hash marks? Or is it starting at the end of the grey portion, and ending with the C-terminal domain. Can the amino acid numbers be added to the beginning and end of the cutout region.

Fig 1 was reworked to better illustrate the p53 regions.

---

## [Decision Letter · Decision Letter 1]

17 Mar 2025

Correlation of p53 oligomeric status and its subcellular localization in the presence of the AML-associated NPM mutant

PONE-D-24-49808R1

Dear Dr. Brodská,

We’re pleased to inform you that your manuscript has been judged scientifically suitable for publication and will be formally accepted for publication once it meets all outstanding technical requirements.

Kind regards,

Swati Palit Deb

Academic Editor

PLOS ONE

Additional Editor Comments (optional):

Reviewers' comments:

Reviewer's Responses to Questions

**Comments to the Author**

1. If the authors have adequately addressed your comments raised in a previous round of review and you feel that this manuscript is now acceptable for publication, you may indicate that here to bypass the “Comments to the Author” section, enter your conflict of interest statement in the “Confidential to Editor” section, and submit your "Accept" recommendation.

Reviewer #2: All comments have been addressed

2. Is the manuscript technically sound, and do the data support the conclusions?

Reviewer #2: Yes

3. Has the statistical analysis been performed appropriately and rigorously? 

Reviewer #2: Yes

4. Have the authors made all data underlying the findings in their manuscript fully available?

Reviewer #2: Yes

5. Is the manuscript presented in an intelligible fashion and written in standard English?

Reviewer #2: Yes

6. Review Comments to the Author

Reviewer #2: The authors have satisfactory addressed all the comets that I have made. All comments have been addressed.

7. PLOS authors have the option to publish the peer review history of their article (what does this mean? ). If published, this will include your full peer review and any attached files.

**Do you want your identity to be public for this peer review?** For information about this choice, including consent withdrawal, please see our Privacy Policy .

Reviewer #2: No

---

## [Editor Report · Acceptance letter]

PONE-D-24-49808R1

PLOS ONE

Dear Dr. Brodská,

I'm pleased to inform you that your manuscript has been deemed suitable for publication in PLOS ONE. Congratulations! Your manuscript is now being handed over to our production team.

Kind regards,

on behalf of

Dr. Swati Palit Deb

Academic Editor

PLOS ONE